# Aetiology and impact of bacterial bloodstream infections in mechanically ventilated COVID-19 patients: A prospective Swedish multicenter cohort study

Isak Olsson[1,2]*, Anna C. Nilsson[3,4], Ingrid Didriksson[1,5], Attila Frigyesi[1,6], Hans Friberg[1,5], Anton Reepalu[3,4], Martin Spångfors[1,7]

**1** Department of Clinical Sciences, Anaesthesiology and Intensive Care, Lund University, Lund, Sweden, **2** Karlstad Hospital, Karlstad, Sweden, **3** Department of Infectious Diseases, Skåne University Hospital, Malmö, Sweden, **4** Department of Translational Medicine, Lund University, Malmö, Sweden, **5** Intensive and Perioperative Care, Skåne University Hospital, Malmö, Sweden, **6** Intensive and Perioperative Care, Skåne University Hospital, Lund, Sweden, **7** Department of Anaesthesia and Intensive Care, Skåne University Hospital, Kristianstad, Sweden

* isak.olsson@med.lu.se

## Abstract

### Objectives

Critically ill COVID-19 patients admitted to the intensive care unit (ICU) are at an increased risk of acquiring bacterial bloodstream infections (BSI). We aimed to describe patient characteristics, risk factors, and the microbiological spectrum in blood cultures and evaluate the impact of ICU-acquired BSI on outcomes in a Nordic setting.

### Methods

A prospective multicenter cohort study was conducted on adult invasively mechanically ventilated (IMV) COVID-19 patients. The primary aim was to identify the proportion of ICU-acquired BSI and its aetiology. Secondary outcomes were duration of IMV, length of stay (LOS), and mortality for individuals with and without BSI, respectively. Logistic regression was used to identify potential predictors of ICU-acquired BSI. Predictors were assessed by calculating an Area Under the Receiver Operating Characteristics (AUROC) curve.

### Results

Of 354 included patients, 17% had an ICU-acquired BSI. *Staphylococcus aureus* was the most common pathogen. Patients with BSI had a longer duration of IMV (20 days versus 9 days, *p < 0.001*), longer ICU-LOS (24 days versus 11 days, *p < 0.001*), and hospital-LOS (38 days versus 24 days, *p < 0.001*). A BSI was associated with

**Data availability statement:** Data availability statement Public access to this data is restricted by the Swedish Authorities (Public Access to Information and Secrecy Act (https://www.government.se/information-material/2009/09/public-access-to-information-and-secrecy-act/). This is due to legal restrictions and that the data contain potentially identifying and sensitive patient information. However, data can be made available for research after a special review that includes approval of the research project by both an ethics committee and the authorities' data safety committee. Data access queries are referred to the Swedish Ethical Review Authority (https://etikprovning-smyndigheten.se/) and KVB, the department responsible for patient data disclosures in Region Skåne, Sweden (https://www.skane.se/om-region-skane/forskning/for-dig-som-forskar/personuppgifter-och-patientdata/kvb-ansokan-for-utlamnande-av-patientdata/). Researchers can contact the Swedish Ethical Review Authority and request access to the data by sending an email to registrator@etikprovning.se.

**Funding:** The author(s) disclosed receipt of the following financial support for the research, authorship, and publication of this article: HF: The Swedish National Health Service (ALF), 2022-0226. Regional funding from Region Skåne, and the Swedish Heart-Lung Foundation, 20210233 and 21023322, Skåne University Hospital grants, and Hans-Gabriel and Alice Trolle-Wachtmeisters Foundation for Medical Research. AF: Regional research support, Region Skåne #2022-1284; Governmental funding of clinical research within the Swedish National Health Service (ALF) #2022:YF0009 and #2022-0075; Crafoord Foundation grant number #2021-0833; Lions Skåne research grants; Skåne University Hospital grants; Swedish Heart and Lund Foundation (HLF) #2022-0352 and #2022-0458. AR holds a 4-year clinical research position, University healthcare unit, Region Skåne, Sweden (2023-099) Funding parties had no part in the study's design, collection of materials, data analysis, or results reporting.

**Competing interests:** The authors have declared that no competing interests exist.

increased mortality; odds ratio (OR) 3.21, 95% CI: 1.61–6.38, $p < 0.001$. Adjusted analyses showed that higher BMI; OR 1.06, 95% CI: 1.01–1.11, $p = 0.014$, diabetes mellitus with organ complications; OR 2.66, 95% CI: 1.33–5.29, $p = 0.005$, and number of symptomatic days before ICU admission; OR 1.04, 95% CI: 1.01–1.07, $p = 0.008$, were associated with a BSI. The AUROC was 0.66 (95% CI: 0.58–0.74).

## Conclusion

ICU-acquired BSIs were found in 17% of critically ill COVID-19 patients and were associated with a longer duration of IMV and LOS as well as increased mortality. *Staphylococcus aureus* was the dominating pathogen. We found several factors associated with ICU-acquired BSIs at ICU admission. However, their ability to predict BSIs was poor.

## Introduction

Most patients with Coronavirus Disease 19 (COVID-19) have mild to moderate respiratory tract infection symptoms or remain asymptomatic [1–3]. A minority develop severe symptoms and critical disease with acute respiratory distress syndrome (ARDS), requiring intensive care with advanced respiratory support including invasive mechanical ventilation (IMV) [2,4,5]. Vaccines against the SARS-CoV-2 virus have dramatically decreased progress to a life-threatening disease [6]. SARS-CoV-2 remains in circulation in society, continuously infecting the public since neither vaccine-induced immunity nor immunity acquired through infection offers complete protection, particularly among the immunocompromised and other vulnerable populations [7,8].

In general, patients treated in intensive care units (ICU) are at high risk of acquiring infectious complications like pneumonia, urinary tract infection or bloodstream infection (BSI). Common risk factors are IMV, intravenous lines and urinary catheters [9]. Risk factors of infectious complications in hospitalised COVID-19 patients include use of broad-spectrum antibiotics, longer ICU-length of stay (ICU-LOS) and hospital length of stay (hospital-LOS), development of ARDS and duration of IMV [10–12]. Common secondary infections include respiratory tract infections and BSIs [10,13,14]. *Staphylococcus aureus* and *Escherichia coli* are common causes of secondary bacterial BSIs [15]. Bacterial BSIs in COVID-19 patients are associated with worse outcomes, e.g., higher rate of ICU admission, need for IMV and increased mortality [16,17]. A higher frequency of bacterial co-infections has been found in critically ill COVID-19 patients compared to critically ill non-COVID-19 patients [18]. In a Swedish context, one study identified a low rate (6.5%) of positive blood cultures among hospitalised COVID-19 patients [19]. Another study, however, found a higher incidence of hospital-acquired BSIs in COVID-19 patients compared to non-COVID-19 patients [20]. Given the implications of antimicrobial resistance on both acquisition and outcome of BSI, it is important to note that there have been a lower reported rate of antimicrobial resistance in the Nordic countries compared to

the other parts of Europe [21]. The rate of deaths by antimicrobial resistance is also lower in the Nordic countries compared to other countries [22]. There is no consensus regarding which factors increase the risk of concurrent infections in critically ill COVID-19 patients [11,14,16,23].

This study aimed to describe patient characteristics and the microbiological spectrum in blood cultures from critically ill COVID-19 patients receiving IMV in the ICU in a Nordic context. Furthermore, we investigated the impact of ICU-acquired BSIs on the duration of IMV, ICU-LOS, hospital-LOS, and 365-day all-cause mortality. Potential patient-specific predictors associated with an ICU-acquired BSI at ICU admission were explored.

## Materials and methods

### Study design

This study was part of the prospective multicenter cohort study SWECRIT COVID-19, including adult patients (≥18 years old) with laboratory-confirmed critical COVID-19 infection treated at the ICU in any of four participating hospitals in the Skåne region, Sweden [24]. Patients were consecutively included on admission to the ICUs at Helsingborg Hospital, Kristianstad Hospital, Skåne University Hospital in Lund and Malmö between May 11, 2020, and May 10, 2021. The participating hospitals adhered to the same regional recommendations and had equal access to resources, such as alcohol-based hand disinfection, mouthguards, and gloves during the pandemic. The virus strain B.1.1.7 (alpha) prevailed throughout the study period, and the vaccine coverage was low throughout the study period [24]. Patients whose COVID-19 was not the primary cause of ICU admission were excluded. Only patients with ARDS receiving IMV were included in this study.

### Data collection

The data collection of background variables has previously been described in detail [22]. In brief, demographic, radiographic, laboratory data, and medication during the ICU stay were obtained from electronic medical records and a regional quality registry, COVID-IR. Data regarding blood culture sets and culture results were extracted from COVID-IR for each patient from five days before hospital admission until the end of the ICU stay. Furthermore, treatment with antibiotic agents during this time frame was collected through COVID-IR.

### Ethical approval

Written informed consent was obtained at ICU admission. If the patient could not make an informed decision due to the acute medical condition, consent was obtained up to one year after ICU admission. For deceased patients, consent was presumed. Ethical approval was approved by the Swedish Ethical Review Authority (DNR 2020/01955, 2020/03483 & 2020/05233). The manuscript was prepared following the STROBE guidelines for observational studies [25].

## Definitions

The ARDS was defined according to the Berlin definition, which states bilateral opacities on chest radiograph or computed tomography in combination with an arterial oxygen partial pressure ratio to fractional inspired oxygen ($PaO_2/FiO_2$) ≤40 kPa and a positive end-expiratory pressure (PEEP) of ≥5 cm $H_2O$. Participants were stratified into mild, moderate or severe ARDS based on the $PaO_2/FiO_2$ ratio [26].

Blood cultures were drawn based on clinical judgment and suspicion of bloodstream infection. There was no local protocol specific for COVID-19 regarding when to collect blood cultures or when to suspect a bloodstream infection. However, regional guidelines state when a clinician should suspect that a patient has sepsis. If a patient presents with clinical signs of sepsis in line with the Surviving Sepsis Campaign Adult Guidelines [27], then a specific sepsis alarm should be activated. If a patient fulfills these criteria blood cultures are routinely collected [28]. However, there might have been situations when blood cultures were drawn based on based on other information or signs of infection that are not specifically stated in the regional guidelines. The regional recommendations state that four blood culture bottles, two aerobic and two anaerobic samples, with a total blood volume of 40 ml, should be obtained before administering intravenous antibiotics. All cultures were analysed using the BD BACTEC FX blood culture system (Becton, Dickinson and Co., USA) at the regional clinical microbiology laboratory in Lund, Sweden, which is accredited according to the ISO 15189 standard.

Positive blood cultures obtained >48 hours after ICU admission to <48 hours after ICU discharge were defined as ICU-acquired infections [10]. Positive blood cultures obtained 5 days before hospital admission and up to 48 hours after hospital admission were defined as community-acquired BSIs, whilst positive cultures obtained >48 hours after hospital admission but <48 hours after ICU admission were defined as hospital ward-acquired BSIs [15]. Furthermore, ICU-acquired BSIs were divided into early and late, depending on whether the positive culture was obtained ≤7 days after ICU admission or later [29]. If a patient acquired an early and late infection, the patient was only included in the first group, i.e., the early infection subgroup.

In the participating ICU's, the first line of antibiotic treatment used for empirical treatment of ICU patients with Covid-19 and suspected bacterial co-infection was third generation Cephalosporins (mainly Cefotaxime), and the second line of treatment was Carbapenems or Piperacillin-tazobactam.

Coagulase-negative staphylococci, "viridans"-group streptococci, *Bacillus species, Corynebacterium species, Micrococcus species, Cutibacterium acnes* and related species were considered probable skin contaminants in blood samples [30,31]. Probable skin contaminants had to be identified in more than one blood sample drawn on the same occasion to be considered a significant finding [30,31]. See S1 Table for the complete classification of pathogens.

The updated version by Quan et al. (2011) was used to calculate the Charlson Comorbidity Index (CCI) [32].

## Outcome measures

The primary outcome was the proportion and aetiology of ICU-acquired BSIs. The secondary outcomes were duration of IMV, ICU-LOS, hospital-LOS, and ICU-, hospital-, and 365-day all-cause mortality measured from ICU admission for individuals with and without BSI, respectively. Demographical and ICU admission characteristics were explored as potential predictors of an ICU-acquired BSI.

## Statistical analysis

Parametric data were presented as mean and standard deviation (SD), whilst non-parametric data were presented as median and interquartile range (q1-q3). Categorical data were presented as numbers (n) and percentages (%). Nominal data were analysed using the Chi-squared test, parametric data using the t-test, and non-parametric data using the Mann-Whitney U test. Univariable and multivariable logistic regressions were performed to explore the variables' association with acquiring an ICU-acquired BSI and to adjust the 365-day mortality for significant confounders. Before the logistic regressions, variables with missing values were imputed by multiple imputations in SPSS using an iterative

Markov chain Monte Carlo method [33]. The rate of missing data was generally low, ranging from 0–23%, and data were missing at random (S2 Table). Before entering the regression analysis, laboratory values were log-transformed (base 10) due to skewness. In line with the "rule of ten", variables with at least ten events per variable and a *p-value<0.2* in univariable analysis were entered in a single step in the multivariable logistic regressions after ruling out multicollinearity by testing for strong correlations with Spearmans- and Point-Biserial-test. [34]. The Hosmer-Lemeshow test was performed to evaluate the goodness of fit in the regression model, and a *p-value>0.05* was considered a good fit [35]. Results of the logistic regressions are presented as unadjusted (OR) and adjusted Odds Ratio (aOR) with 95% Confidence Interval (CI), respectively.

The prediction was assessed by creating receiver operating characteristic (ROC) curves from the logistic regression models, and Areas Under the Receiver Operating Characteristics (AUROC) were calculated.

A sensitivity analysis was performed by comparing the results of the multivariable logistic regressions using the imputed and original data sets with missing variables included (S3 Table). Statistical significance was set to *p<0.05,* and statistical analyses were conducted using IBM SPSS version 28.0.0.0.

## Results

### General patient characteristics

A total of 354 patients were included (Fig 1). The median age was 66 (57–73) years, 73% were male, and the median body mass index (BMI) was 30 (27–35) kg/m². The median number of symptomatic days before ICU admission was 11 (8–15) days, the median Sequential Organ Failure Assessment (SOFA) score at ICU admission was 8 (6–10) points, and the median Simplified Acute Physiology Score 3 (SAPS 3) was 63 (54–71) points. On day 2 of IMV, 7% had mild ARDS, 60% moderate ARDS and 33% severe ARDS. Of the included patients, 98% received antibiotic therapy at some point during their ICU stay. Blood cultures were drawn from 99% of patients at least once during the hospital stay and from 78% during the ICU stay. The ICU mortality was 37%, hospital mortality was 44%, and 365-day mortality was 45% (Table 1).

### Bloodstream infections

In all, 12 (3%) patients had a community-acquired BSI, 8 (2%) patients had a hospital ward-acquired BSI, and 58 (17%) patients had an ICU-acquired BSI. Patients with an ICU-acquired BSI had higher BMI, 32 (28–38) kg/m² versus 30 (27–35) kg/m² (*p=0.028*), than those who did not acquire BSI. (Table 1).

Fig 2 shows the most frequently identified pathogens causing ICU-acquired BSIs. *Staphylococcus aureus* (n=23) was the most common pathogen, followed by coagulase-negative staphylococci (n=12), *Enterococcus faecalis* (n=8), and *Enterococcus faecium* (n=8).

A multivariable logistic regression, including variables known at ICU admission, showed that higher BMI (aOR 1.06, 95% CI: 1.01–1.11, *p=0.014*), diabetes mellitus with signs of organ complications (aOR 2.66, 95% CI: 1.33–5.29, *p=0.005*) and the number of symptomatic days before ICU admission (aOR 1.04, 95% CI: 1.01–1.07, *p=0.008*) were significantly associated with a BSI in the ICU (Table 2). The AUROC was 0.66 (95% CI: 0.58–0.74). Sensitivity analysis showed no significant differences from the regression analysis with imputed data.

Categorising the 58 patients with an ICU-acquired BSI showed that 79% had a late ICU-acquired BSI, 21% had an early ICU-acquired BSI, and one patient had both an early and late ICU-acquired BSI. The subgroup of patients with a late infection had significantly worse severity of ARDS (*p=0.039*). See Table 3 for a detailed description of patient characteristics stratified by early versus late ICU-acquired BSI.

In all, the 58 patients with an ICU-acquired BSI had a longer duration of IMV (20 days versus 9 days, *p<0.001*), length-of-ICU stay (24 days versus 11 days, *p<0.001*), and length-of-hospital stay (38 days versus 24 days, *p<0.001*). Mortality rates were higher, with a 365-day mortality rate of 60% compared to 42% for those who did not acquire a BSI in the ICU

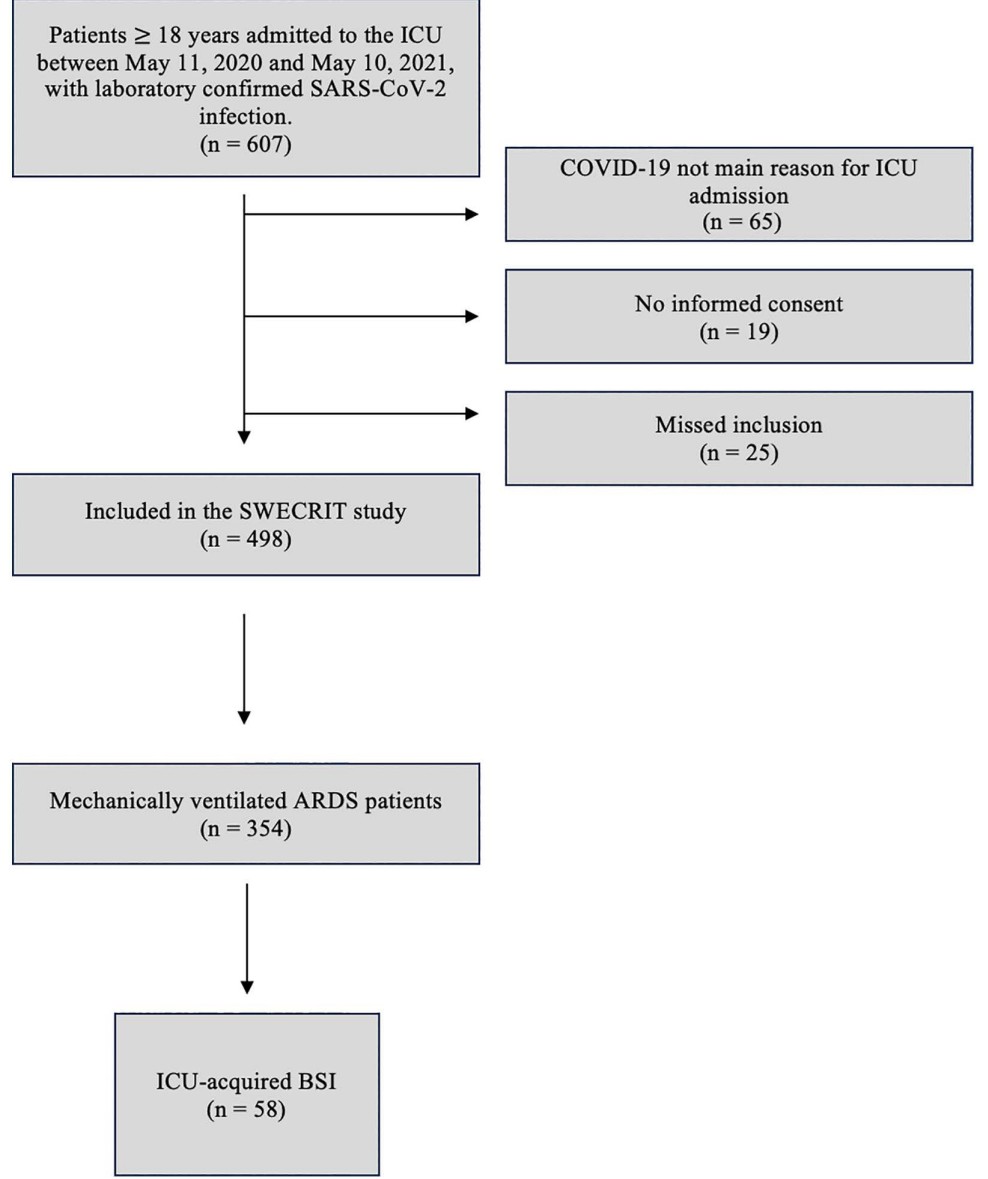

**Fig 1. Flowchart of the 354 COVID-19 patients with ARDS included in the study.** Abbreviations: ARDS: Acute respiratory distress syndrome, BSI: Bloodstream infection, ICU: Intensive care unit.

($p = 0.01$) (Table 1). The association between 365-day mortality and BSI remained after adjusting for age, BMI, Clinical Frailty Scale (CFS), CCI, SAPS 3 and $PaO_2/FiO_2$-ratio at day 2 of IMV in a multivariable regression analysis (aOR 3.21, 95% CI: 1.61–6.38, $p < 0.001$) (S4 Table).

## Discussion

This study aimed to investigate the bacterial aetiology and clinical consequences of ICU-acquired BSIs among patients with critical COVID-19 who required IMV. Our main finding was that 17% of the included patients had an ICU-acquired BSI, the majority occurring more than one week after ICU admission. *Staphylococcus aureus* was the most common

**Table 1. Baseline characteristics, clinical interventions and outcomes in 354 critically ill COVID-19 patients treated with invasive mechanical ventilation, stratified by an ICU-acquired bacterial bloodstream infection.**

| | Overall (n = 354) | No ICU acquired BSI (n = 296) | ICU acquired BSI (n = 58) | P-value |
|---|---|---|---|---|
| **Demographics** | | | | |
| Age, years | 66 (57 −73) | 66 (57 - 73) | 66 (58 - 70) | 0.628 |
| Male | 72.9% | 72.3% | 75.9% | 0.577 |
| BMI, kg/m^2 | 30.0 (26.9 - 35.1) | 29.6 (26.7 - 34.7) | 31.9 (28.4 - 37.5) | **0.028** |
| Clinical frailty scale, score | 3.0 (2.0 - 3.5) | 3.0 (2.0 - 4.0) | 3.0 (2.0 - 3.0) | 0.429 |
| Hypertension | 56.0% | 54.6% | 63.2% | 0.234 |
| Ever smoker | 43.2% | 41.9% | 50.0% | 0.254 |
| Charlson Comorbidity Index | | | | |
| CCI, score | 3.0 (2.0-4.0) | 3.0 (2.0-4.0) | 3.0 (2.0-4.0) | 0.165 |
| Chronic pulmonary disease | 19.5% | 19.3% | 20.7% | 0.801 |
| Rheumatologic disease | 9.0% | 9.8% | 5.2% | 0.261 |
| Liver disease | 1.7% | 1.4% | 3.4% | 0.256 |
| Diabetes | 32.5% | 31.1% | 39.7% | 0.202 |
| Diabetes with organ complications | 15.3% | 14.0% | 29.6% | **0.004** |
| Malignancy | 10.2% | 9.5% | 13.8% | 0.318 |
| Moderate to severe renal disease | 4.0% | 3.7% | 5.2% | 0.710 |
| Medication prior to admission to hospital | | | | |
| Chronic steroid therapy | 5.6% | 5.7% | 5.2% | 1.000 |
| Other immunosuppressive agents | 9.0% | 8.8% | 10.3% | 0.705 |
| **ICU admission characteristics** | | | | |
| Community acquired BSI | 3.4% | 3.0% | 5.2% | 0.424 |
| Hospital acquired BSI | 2.3% | 2.4% | 1.7% | 1.000 |
| Symptoms prior to ICU admission, days | 11 (8 −15) | 11 (8 −15) | 12 (7 - 18) | 0.195 |
| Antibiotic treatment at hospital admission | 47.7% | 49.0% | 41.4% | 0.289 |
| Laboratory testing at ICU admission | | | | |
| Creatinine, µmol/L | 85 (68 - 130) | 84 (68 - 129) | 89 (66 - 135) | 0.796 |
| Leukocytes, 10^9/L | 11.0 (8.2 - 15.7) | 11.0 (8.3 - 15.6) | 11.0 (8.1 - 16.7) | 0.852 |
| Neutrophils, 10^9/L | 9.1 (6.5 - 13.0) | 9.1 (6.6 - 12.7) | 9.3 (6.1 - 14.2) | 0.940 |
| Thrombocytes, 10^9/L | 272 (207 - 365) | 274 (211 - 367) | 271 (195 - 361) | 0.502 |
| CRP, mg/L | 158.50 (94.00 - 241.75) | 156 (94 - 245) | 169 (85 - 231) | 0.925 |
| Procalcitonin, µg/L | 0.500 (0.235 - 1.300) | 0.530 (0.245 - 1.400) | 0.385 (0.183 - 0.920) | 0.135 |
| Lactate, mmol/L | 2.250 (1.800 - 3.025) | 2.3 (1.8 - 3.1) | 2.1 (1.8 - 2.7) | 0.230 |
| Bilirubin, µmol/L | 10 (7 - 14) | 10 (7 - 14) | 10 (6 - 14) | 0.926 |
| D-dimer, mg/L FEU | 2.2 (1.1 - 7.9) | 2.2 (1.1 - 8.0) | 2.5 (0.9 - 7.7) | 0.708 |
| Ferritin, µg/L | 1420 (878 - 2344) | 1430 (902 - 2392) | 1159 (603 - 2213) | 0.161 |
| IL-6, ng/L | 130 (52 - 299) | 137 (57 - 288) | 102 (44 - 382) | 0.499 |
| SOFA score at ICU admission | 8 (6 - 10) | 8 (6 - 9) | 9 (4 - 11) | 0.331 |
| SAPS 3 | 63 (54 - 71) | 63 (54 - 71) | 59 (50 - 69) | **0.112** |
| **Clinical variables & interventions** | | | | |
| PaO$_2$/FiO$_2$ ratio at day 2 of intubation, kPa | 16 (12 - 20) | 16 (12 - 20) | 15 (12 - 20) | 0.462 |
| Severity of ARDS at day 2 of intubation | | | | 1.000 |
| Mild ARDS | 7.3% | 7.4% | 6.9% | |
| Moderate ARDS | 59.3% | 59.1% | 60.3% | |
| Severe ARDS | 33.3% | 33.4% | 32.8% | |
| Antibiotic treatment during ICU stay | 98.0% | 97.6% | 100% | 0.237 |

*(Continued)*

**Table 1.** (Continued)

|  | Overall (n = 354) | No ICU acquired BSI (n = 296) | ICU acquired BSI (n = 58) | P-value |
|---|---|---|---|---|
| Antiviral treatment during ICU stay | 13.3% | 13.5% | 12.1% | 0.767 |
| Corticosteroid treatment during ICU stay | 97.2% | 97.0% | 98.3% | 0.580 |
| **Outcomes** | | | | |
| IMV, days | 9.9 (5.44 - 19.3) | 9.0 (4.9 - 15.3) | 20.3 (12.4 - 28.7) | **<0.001** |
| Length of ICU stay, days | 12.67 (7.68 - 23.16) | 11.4 (7.1 - 19.9) | 23.6 (16.3 - 35.1) | **<0.001** |
| Length of hospital stay, days | 25.9 (18.4 - 50.6) | 24.4 (17.2 - 47.2) | 38.3 (23.4 - 72.0) | **<0.001** |
| ICU mortality | 37.3% | 34.1% | 53.4% | **0.005** |
| Hospital mortality | 43.5% | 40.5% | 58.6% | **0.011** |
| 365-day mortality | 44.9% | 41.9% | 60.3% | **0.010** |

Data are presented as percentages (%) or medians and interquartile ranges. Abbreviations: ARDS: Acute respiratory distress syndrome, BMI: Body mass index, BSI: Bacterial bloodstream infection, CCI: Charlson Comorbidity Index, CRP: C-reactive protein, ICU: Intensive care unit, IL-6: Interleukin-6, IMV: Invasive mechanical ventilation, IQR: Interquartile range, $PaO_2/FiO_2$: Partial pressure of arterial oxygen to fraction of inspired oxygen, SAPS 3: Simplified acute physiology Score 3, SOFA: Sequential organ failure assessment.

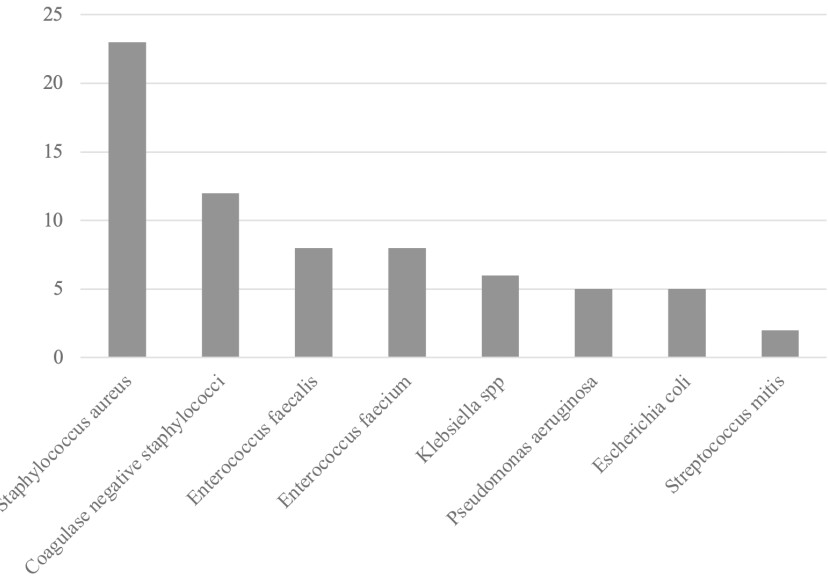

**Fig 2. Bar chart illustrating the frequency of bacteria causing ICU-acquired bloodstream infections in 354 critically ill COVID-19 patients treated with invasive mechanical ventilation.** Only bacterial species occurring in more than one patient are included in the chart. Abbreviation: ICU: Intensive care unit.

pathogen. An ICU-acquired BSI was independently associated with more than threefold odds of dying within 365 days as well as a longer duration of IMV and longer ICU-LOS.

The incidence of ICU-acquired BSIs in our study is in line with the findings in two large multicenter studies [18,36]. In contrast, two other European studies show that up to one-third of patients acquire a BSI in the ICU [10,37]. One of these studies, however, was a small single-centre study [37], making it less relevant to compare with our larger multi-centre study. We therefore suggest that the rate of ICU-acquired BSIs in this patient population can be estimated to range between 15%−30%.

**Table 2. Univariable and multivariable logistic regression for the associations with ICU-acquired bacterial bloodstream infections in 354 critically ill COVID-19 patients treated with invasive mechanical ventilation.**

| Univariable logistic regression | Odds Ratio (OR) | 95% Confidence Interval (CI) | P-value |
|---|---|---|---|
| BMI | 1.050 | 1.006 - 1.095 | *0.025* |
| Charlson Comorbidity Index (CCI) | | | |
| CCI score | 1.052 | 0.907 - 1.221 | *0.499* |
| Diabetes with signs of organ complications | 2.586 | 1.325 - 5.049 | *0.005* |
| Duration of symptoms prior to ICU admission | 1.030 | 1.002 - 1.059 | *0.039* |
| Antibiotic treatment at hospital admission | 0.735 | 0.416 - 1.300 | *0.290* |
| Laboratory testing at ICU admission | | | |
| Log 10 Procalcitonin | 0.697 | 0.468 - 1.038 | *0.076* |
| Log 10 Ferritin | 0.637 | 0.322 - 1.259 | *0.195* |
| SAPS 3 | 0.983 | 0.961 - 1.006 | *0.150* |
| **Final adjusted model of the multivariable logistic regression** | **Adjusted Odds Ratio (aOR)** | **95% Confidence Interval (CI)** | *P-value* |
| BMI | 1.058 | 1.012 - 1.107 | *0.014* |
| Diabetes with complications | 2.657 | 1.334 - 5.292 | *0.005* |
| Duration of symptoms prior to ICU admission | 1.041 | 1.010 - 1.072 | *0.008* |
| AUROC: 0.66 (95% CI = 0.58–0.74, *p < 0.001*) | | | |
| Hosmer-Lemeshow test for goodness of fit: *P = 0.94* | | | |

Abbreviations: aOR: Adjusted Odds Ratio, AUROC: Area Under the Receiver Operating Characteristics, BMI: Body mass index, BSI: Bacterial bloodstream infection, CCI: Charlson Comorbidity Index, CI: Confidence Interval, ICU: Intensive care unit, Log10: Base 10 logarithm, OR: Odds Ratio and SAPS 3: Simplified Acute Physiology Score 3.

This multicenter study was performed in a Nordic context. A Finnish study identified positive blood culture rates of close to 6% among COVID-19 patients treated in the ICU. In comparison, Swedish study identified positive blood cultures of approximately 7% among all patients treated in the hospital with COVID-19 [19,38]. These are numbers far below our rate of 17% of ICU-acquired BSIs. The Finnish ICU study, however, included only 132 patients, and it is unclear to what extent they received IMV. Our study included a larger sample, and all patients received IMV, which may explain the difference in incidence rate since critically ill patients receiving IMV are more susceptible to concomitant BSIs [23]. The definition of significant growth versus possible contamination may also differ from ours, possibly affecting the results [19].

## Microbiological spectra

We identified *Staphylococcus aureus* as the dominating microbiological agent, a well-known cause of ICU-acquired BSIs [10,15,18]. Previous studies have found equally high or higher frequencies of species like *Klebsiella pneumoniae,* coagulase-negative staphylococci, *Enterococcus species. Escherichia coli* and *Pseudomonas aeruginosa* [15,18,36]. Studies investigating ICU-acquired BSIs vary regarding the cause of ICU admission, treatments and interventions during ICU stay, as well as regional differences in microbiological spectra, antimicrobial resistance patterns and deaths associated with antimicrobial resistance. Hence, it is difficult to draw any conclusions on the dominating pathogens in other settings [10,15,18,21,22,36]. Of note, the second to fourth most common causative agents behind BSI in our study were organisms frequently harboring resistance mechanisms conferring reduced susceptibility or resistance to third generation cephalosporins, the first-line empiric antibiotic used at the participating ICUs [39,40]. With this in mind, we consider it essential to emphasize that this study provides valuable insights into the Nordic context in general and the Swedish context in particular.

**Table 3. Baseline characteristics, clinical interventions and outcomes of 58 critically ill COVID-19 patients treated with invasive mechanical ventilation with an ICU-acquired bacterial bloodstream infection, stratified by early (≤ 7 days after ICU admission) or late (≥ 7 days after ICU admission) infection.**

| | Early ICU acquired BSI (n = 12) | Late ICU acquired BSI (n = 46) | P-value |
|---|---|---|---|
| **Demographics** | | | |
| Age, years | 66 (57 - 72) | 66 (58 - 70) | 0.885 |
| Male | 83.3% | 73.9% | 0.711 |
| BMI, kg/m^2 | 28.6 (25.5 - 38.4) | 32.2 (29.9 - 37.5) | 0.272 |
| Clinical frailty scale, score | 3 (2 - 3) | 3 (2 - 3) | 0.952 |
| Hypertension | 50.0% | 66.7% | 0.327 |
| Ever smoker | 58.3% | 47.8% | 0.517 |
| Charlson Comorbidity Index (CCI) | | | |
| CCI, score | 3.5 (2.0 - 4.0) | 3.0 (2.0 - 4.0) | 0.739 |
| Chronic pulmonary disease | 33.3% | 17.4% | 0.247 |
| Rheumatologic disease | 0% | 6.5% | 1.000 |
| Liver disease | 0% | 4.3% | 1.000 |
| Diabetes | 25.0% | 43.5% | 0.329 |
| Diabetes with organ complications | 25.0% | 28.3% | 1.000 |
| Malignancy | 8.3% | 15.2% | 1.000 |
| Moderate to severe renal disease | 8.3% | 4.3% | 0.509 |
| Medication prior to admission to hospital | | | |
| Chronic steroid therapy | 0% | 6.5% | 1.000 |
| Other Immunosuppressive agents | 0% | 13.0% | 0.328 |
| **ICU admission characteristics** | | | |
| Community acquired BSI | 16.7% | 2.2% | 0.106 |
| Hospital acquired BSI | 0% | 2.2% | 1.000 |
| Symptoms prior to ICU admission, days | 11 (8 - 17) | 13 (7 - 18) | 0.569 |
| Antibiotic treatment at hospital admission | 25.0% | 45.7% | 0.324 |
| Laboratory testing at ICU admission | | | |
| Creatinine, µmol/L | 68 (60 - 132) | 94 (71 - 135) | 0.207 |
| Leukocytes, 10^9/L | 10.9 (5.9 - 16.6) | 11.0 (8.2 - 17.7) | 0.478 |
| Neutrophils, 10^9/L | 9.3 (4.0 - 13.8) | 9.2 (6.2 - 15.4) | 0.448 |
| Thrombocytes, 10^9/L | 261 (186 - 275) | 286 (198 - 372) | 0.192 |
| CRP, mg/L | 138 (61 - 234) | 175 (85 - 231) | 0.578 |
| Procalcitonin, µg/L | 0.340 (0.130 - 0.900) | 0.390 (0.185 - 1.110) | 0.710 |
| Lactate, mmol/L | 2.0 (1.5 - 2.5) | 2.2 (1.9 - 2.7) | 0.260 |
| Bilirubin, µmol/L | 10 (6 - 17) | 10 (6 - 14) | 0.840 |
| D-dimer, mg/L FEU | 2.5 (0.5 - 35.0) | 2.5 (0.9 - 7.7) | 0.861 |
| Ferritin, µg/L | 1460 (1008 - 2414) | 1159 (572 - 2202) | 0.377 |
| IL-6, ng/L | 105 (44 - 416) | 102 (40 - 349) | 0.647 |
| SOFA score at ICU admission | 9 (4 - 11) | 9 (4 - 11) | 0.883 |
| SAPS 3 | 66 (51 - 73) | 59 (50 - 67) | 0.241 |
| **Clinical variables & interventions** | | | |
| PaO$_2$/FiO$_2$ ratio at day 2 of intubation, kPa | 19 (12 - 25) | 15 (12 - 18) | 0.210 |
| Severity of ARDS at day 2 of intubation | | | ***0.039*** |
| Mild ARDS | 25.0% | 2.2% | |
| Moderate ARDS | 50.0% | 63.0% | |
| Severe ARDS | 25.0% | 34.8% | |

*(Continued)*

**Table 3.** (Continued)

| | Early ICU acquired BSI (n = 12) | Late ICU acquired BSI (n = 46) | P-value |
|---|---|---|---|
| Antibiotic treatment during ICU stay | 100% | 100% | 1.000 |
| Antiviral treatment during ICU stay | 8.3% | 13.0% | 0.656 |
| Corticosteroid treatment during ICU stay | 100% | 97.8% | 0.606 |
| **Outcomes** | | | |
| IMV, days | 17.4 (9.6 - 34.3) | 20.8 (14.1 - 27.6) | 0.578 |
| Length of ICU stay, days | 19.2 (12.5 - 41.5) | 24.0 (16.9 - 35.1) | 0.388 |
| Length of hospital stay, days | 28.7 (16.2 - 44.5) | 42.7 (24.4 - 72.4) | 0.120 |
| ICU mortality | 50.0% | 54.3% | 0.788 |
| Hospital mortality | 58.3% | 58.7% | 0.982 |
| 365-day mortality | 58.3% | 60.9% | 0.873 |

Data are presented as percentages (%) or medians and interquartile ranges. Abbreviations: ARDS: Acute respiratory distress syndrome, BMI: Body mass index, BSI: Bacterial bloodstream infection, CCI: Charlson Comorbidity Index, CRP: C-reactive protein, ICU: Intensive care unit, IL-6: Interleukin-6, IMV: Invasive mechanical ventilation, IQR: Interquartile range, $PaO_2/FiO_2$: Partial pressure of arterial oxygen to fraction of inspired oxygen, SAPS 3: Simplified acute physiology Score 3, SOFA: Sequential organ failure assessment.

## Early versus late ICU-acquired bloodstream infections

Almost 80% of the ICU-acquired BSIs in the present study occurred late (more than 7 days) after ICU admission, which aligns with previous findings in COVID-19 patients [23]. Late infections in the ICU are not only specific to patients with COVID-19; the large EUROBACT-2 study showed that late ICU-acquired BSIs were more common among general ICU patients [29].

## Outcomes and risk factors

An ICU-acquired BSI was associated with a more than twice as long duration of IMV and ICU-LOS and close to twice as long hospital-LOS compared to those not acquiring a BSI, which is similar to what has previously been reported [10,11,37]. It is worth highlighting that these differences were seen when comparing two groups of patients with no significant differences in variables like C-reactive protein (CRP), SOFA score and SAPS3 score at ICU admission as well as severity of ARDS at day 2 of intubation. With the granularity of our data we cannot conclude whether a secondary BSI in our study significantly increased the hospital- and ICU-LOS or if the contrary is true, i.e., that the prolonged hospital- and ICU-LOS increased the risk of developing a BSI, which has been shown in previous studies [23,29,36,41]. However, it was out of the scope of the current study to further analyse possible causal relationships.

In the adjusted analyses, including variables at ICU admission, we found that a higher BMI, diabetes mellitus with organ complications, and the number of symptomatic days before ICU admission were risk factors associated with an ICU-acquired BSI. Obesity has, in line with our previous findings, been identified as a risk factor for BSIs in critically ill COVID-19 patients [23]. Diabetes mellitus has also been associated with secondary infections in critically ill COVID-19 patients [37]. However, considering the limited predictive ability of our regression model (AUROC = 0.66), it remains difficult to predict which patients will have an ICU-acquired BSI using the chosen pre-ICU admission risk factors [42].

The study aimed to investigate patient-specific risk factors associated with an ICU-acquired BSI. We acknowledge that there exist other potential risk factors such as the use of central venous lines and external risk factors like to what extent clinical staff adhered to infection control measures like basic hand hygiene routines. Exploring those non-patient-specific risk factors was beyond the scope of the current study, but they would be interesting to further investigate and explore in future studies.

In the present study, an ICU-acquired BSI was associated with a more than threefold increase in the odds of dying within 365 days of ICU admission, even when adjusting for confounders. This is in line with previous studies where an ICU-acquired BSI has been associated with a twofold or more increase in the odds of dying among critically ill COVID-19 patients [43–45]. Importantly, although we found a significant association between mortality and ICU-acquired BSI, the results must be interpreted with caution. We cannot rule out that some patients acquiring a BSI during the ICU stay were more severely ill, with an expected increased mortality rate even without a secondary BSI.

### Strengths and limitations

This study's major strength is the large population of prospectively included consecutive, critically ill COVID-19 patients receiving IMV from multiple centres over 12 months. The rate of missed inclusions and patients excluded due to not receiving consent was very low. Furthermore, the rate of missing data was low, generating more robust results.

Our study's limitations include its observational design, which does not allow for causal conclusions but merely associations. The protocol did not include systematic blood cultures before antibiotic treatment in the ICU. Yet, the routinely obtained blood culture rate was high, minimising the risk of misclassified BSIs. Furthermore, the study design did not allow for exploring temporal associations between ICU-LOS and BSI. No adjustments for multiple testing were made, and the results should only be considered hypothesis-generating due to the exploratory design. Additionally, the depth and details of the data regarding information about the administration of antibiotics could have been more detailed than what was collected for this study. For example, administration of antibiotic agents could have been collected as a continuous variable, allowing for temporal comparisons. Hence, we consider this a potential limiting factor in this study.

### Conclusion

Secondary ICU-acquired BSIs are common, occurring in 17% of critically ill COVID-19 patients receiving IMV. Most infections occurred more than one week after ICU admission, and the most common pathogen was *Staphylococcus aureus.* An ICU-acquired BSI was associated with more than threefold odds of dying within 365 days as well as a longer duration of IMV and longer ICU-LOS. We found several factors associated with ICU-acquired BSIs at ICU admission. However, their ability to predict BSIs was poor.

### Supporting information

**S1 Table. Bacteria occurring in blood cultures in the study, stratified by clinical significance.**
(DOCX)

**S2 Table. Missing data.** Data presented as numbers (percentages). Abbreviations: Percentages: (%), ARDS: Acute respiratory distress syndrome, BMI: Body mass index, BSI: Bacterial bloodstream infection, CCI: Charlson Comorbidity Index, CRP: C-reactive protein, ICU: Intensive care unit, IL-6: Interleukin-6, IMV: Invasive mechanical ventilation, SAPS 3: Simplified acute physiology score 3, SOFA: Sequential organ failure assessment, IQR: Interquartile range, $PaO_2/FiO_2$: Partial pressure of arterial oxygen to fraction of inspired oxygen, SAPS 3: Simplified acute physiology score 3, SOFA: Sequential organ failure assessment.
(DOCX)

**S3 Table. Sensitivity analysis comparing the results of the multivariable logistic regression for the associations with ICU-acquired bacterial bloodstream infection between the original data set with missing variables and the imputed data set.** Abbreviations: aOR: Adjusted Odds Ratio, BSI: Bacterial bloodstream infection, BMI: Body mass index, CI: Confidence Interval, ICU: Intensive care unit.
(DOCX)

**S4 Table. Univariable and multivariable logistic regressions for the associations with 365-day mortality rate in 354 critically ill COVID-19 patients treated with invasive mechanical ventilation.** Supplementary Table 4 shows the univariable logistic regression result, including acquiring an ICU-acquired BSI and variables known at ICU admission and their association with 365-day mortality. Data are presented as p-value, Odds Ratio and 95% Confidence Interval. P--values < 0.05 were considered significant. Covariates included in the multivariable logistic regression were age, BMI, CCI, CFS, duration of symptoms before ICU admission, history of smoking (ever smoker), ICU-acquired BSI, medication with immunosuppressive agents before hospitalisation, $PaO_2/FiO_2$ ratio at day 2 of intubation and SAPS 3. Abbreviations: aOR: Adjusted Odds Ratio, AUROC: Area Under the Receiver Operating Characteristics BMI: Body mass index, BSI: Blood-stream infection, CCI: Charlson Comorbidity Index, CI: Confidence Interval, ICU: Intensive care unit, OR: Odds Ratio, $PaO_2/FiO_2$: Partial pressure of arterial oxygen to fraction of inspired oxygen, SAPS 3: Simplified Acute Physiology Score 3 and SOFA: Sequential Organ Failure Assessment.
(DOCX)

## Acknowledgments

We acknowledge all the participating patients, their next-of-kin, and the staff at the participating ICUs of Skåne University Hospital in Malmö and Lund, Helsingborg Hospital, and Kristianstad Hospital for their invaluable contributions to the study.

## Author contributions

**Conceptualization:** Isak Olsson, Anna C. Nilsson, Ingrid Didriksson, Attila Frigyesi, Hans Friberg, Anton Reepalu, Martin Spångfors.

**Data curation:** Ingrid Didriksson.

**Formal analysis:** Isak Olsson, Anna C. Nilsson, Martin Spångfors.

**Funding acquisition:** Attila Frigyesi, Hans Friberg.

**Investigation:** Isak Olsson, Ingrid Didriksson, Anton Reepalu, Martin Spångfors.

**Methodology:** Isak Olsson, Anna C. Nilsson, Attila Frigyesi, Hans Friberg, Anton Reepalu, Martin Spångfors.

**Project administration:** Ingrid Didriksson, Hans Friberg.

**Supervision:** Martin Spångfors.

**Visualization:** Isak Olsson, Martin Spångfors.

**Writing – original draft:** Isak Olsson.

**Writing – review & editing:** Anna C. Nilsson, Ingrid Didriksson, Attila Frigyesi, Hans Friberg, Anton Reepalu, Martin Spångfors.

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
