## [Decision Letter · Decision Letter 0]

10 Mar 2025

Dear Dr. Olsson,

We look forward to receiving your revised manuscript.

Kind regards,

Lorenzo Righi

Academic Editor

PLOS ONE

Journal Requirements:

2. In the online submission form, you indicated that data collected for this study, including individual participant data, are not publicly available because they contain potentially identifying and sensitive patient information regulated by Swedish law. Data can be made available upon reasonable request from investigators.

Reviewers' comments:

Reviewer's Responses to Questions

**Comments to the Author**

1. Is the manuscript technically sound, and do the data support the conclusions?

Reviewer #1: Yes

Reviewer #2: Partly

2. Has the statistical analysis been performed appropriately and rigorously?

Reviewer #1: Yes

Reviewer #2: Yes

3. Have the authors made all data underlying the findings in their manuscript fully available?

Reviewer #1: No

Reviewer #2: Yes

4. Is the manuscript presented in an intelligible fashion and written in standard English?

Reviewer #1: Yes

Reviewer #2: Yes

Reviewer #1: During a one-year study period at the beginning of Covid-19 pandemic, Olsson et al. conducted a prospective cohort study among mechanically ventilated Covid-19 patients in six ICUs on four study sites in Sweden. The aims of the study were to describe patient characteristics and the microbiological spectrum in blood cultures, to investigate the impact of ICU-acquired BSI on various outcomes and to explore predictors for ICU-acquired BSI on admission to ICU. Several published studies have already investigated these objectives in various settings. It is not clear from the objectives what the authors hope to learn from the study that is not already known.

In their study, the authors focus exclusively on patient-specific risk factors for BSI acquisition such as BMI, diabetes etc. They do not explore external risk factors such as low staff adherence to basic infection control measures (e.g. hand hygiene, especially before clean/aseptic procedures) which has been shown to influence BSI rates (e.g. see Hansen Carter et al., Infect Control Hosp Epidemiol. 2016). In many hospitals hand hygiene compliance has decreased during the pandemic in favour of wearing personal protective equipment including gloves, with nosocomial infections including central-line associated BSI being a consequence (e.g. see Schlosser et al., BMC Infect Dis 2024).

A prospective multi-center cohort study could have tried to address these issues, at least by exploring and reporting differences between study sites or units in the statistical analyses. If differences between study sites can be observed, what are potential explanations? E.g. did the use of alcohol-based hand rubs or of non-sterile gloves differ? Were hand sanitizers available or was there a shortage during the study period? Was there a different patient to health-care worker ratio (understaffing)?

Specific comments

Methods

-In general, the methods section is very thin, some of the methods only appear later in the results section or remain unclar, see below.

-The choice of primary and secondary outcomes chosen by the authors is somewhat uncommon. Usually secondary outcomes should help interpret the results for the primary outcome. In contrast, in this study the secondary outcome depends on the primary outcome. To make it easier to follow, these should be two different steps in the text as well as in the tables. The classification into early and late BSI is not specified in the methods section. However, I have some reservations about this somewhat arbitrary classification, see results below.

-Lines 153-156, secondary outcome measures: It is not specified whether the duration of IMV and ICU-LOS is measured from ICU admission or from BSI onset, respectively. Same for 365-day mortality, is this from admission, from discharge, from BSI onset?

-The duration of devices present (e.g. central venous line) should be included as a continuous variable in the regression model investigating predictors for BSI, as well as the duration of ICU stay.

-In the introduction the authors mention that Covid-19 is associated with numerous healthcare-associated infections, not only BSI. In order to explore their influence on the long-term clinical outcomes of interest they should also be included in the regression model.

Results

-Table 1: Why were patients with BSI that were not acquired at the ICU not excluded? Assuming they did receive antimicrobial treatment for their BSI that was acquired elsewhere, was their risk of acquiring another BSI really comparable to that of the other patients? The same holds true for patients receiving antibiotic treatment for other infections than BSI, but that would probably decrease your study population by too many patients.

-The authors state that 98% of patients received antimicrobial therapy during their ICU stay, but what was proportion receiving therapy before onset of BSI? This should be adjusted for in the logistic regression model.

-Lines 247-251: The classification of ICU acquired BSI into early and late onset with a cutoff of 7 days seems arbitrary to me. Rather than exploring it as an outcome, the authors should take advantage of this continuous variable and adjust the analysis of the other risk factors for the length of ICU stay prior to BSI onset, see above. I would assume that the risk of acquiring BSI increases with duration of ICU treatment, i.e. time at risk.

-Table 3: Mentioning outcomes measured at the end of the hospital stay/one year after hospital discharge in the same table as the risk factors is difficult to grasp for the reader.

-Figure 2: The Y-Axis title is missing.

Discussion

-Lines 315-318: The authors discuss devices as points of entry of bacteria. However, neither do they explore the presence of these devices and their handling as risk factors for BSI, nor do they discuss the preventability of these infections.

-Lines 322-325: The conclusion is correct that from the results at hand the authors cannot distinguish between chicken and egg. As suggested above they should at least differentiate between exposure and outcome time, i.e. time before and after developing BSI.

-Lines 327-334: The authors indirectly state that their statistical model did not seem to include the relevant predictors for BSI. For improvements to the model see comments above.

Reviewer #2: The authors refers 98% of the study subjects received antibiotic therapy at some point their ICU stay. They do not mention the spectrum of this antibiotics. There is no definition of antibiotic resistance i.e Gram-negative bacteria as Enterobacteriaceae with extended-spectrum β-lactamase production or piperacillin-tazobactam resistance, and Acinetobacter spp and Pseudomonas aeruginosa with piperacillin-tazobactam or carbapenem resistance. In this study Staphylococcus aureus was the most common pathogen identified,usually in this ICU context there is a swab examination. Why did they not collect swabs to evaluate the presence of methicillin-resistant Staphylococcus aureus (MRSA) or vancomycin-resistant Enterococcus (VRE)? The most frequent pathogens reported in ICU, particularly when the aim of the study is to investigate the bacterial aetiology. The results shows 45% of the subjects developed late ICU acquired BSI, they received antibiotics at the hospital admission, it is interesting to know about any antibiotic rotation strategies used during the hospitalization, and the therapy the subjects received previous the hospitalization, with the intention to evaluate pre-ICU admission risk factors.

**Do you want your identity to be public for this peer review?** For information about this choice, including consent withdrawal, please see our Privacy Policy

Reviewer #1: No

Reviewer #2: No

---

## [Author Response · Author response to Decision Letter 1]

12 Sep 2025

Dear reviewers!

We want to thank the editor and reviewers for their wise and insightful comments. We have tried our very best to make adequate changes to the manuscript and we have uploaded a new improved version of the manuscript as well as a copy of the manuscript with trackable highlighted changes. We believe that your comments have made it possible for us to substantially improve the quality of the manuscript.

The response to each comment can be seen in the submitted rebuttal letter named "Response to Reviewers".

On behalf of the co-authors,

Yours sincerely,

Isak Olsson, MD.

---

## [Decision Letter · Decision Letter 1]

15 Oct 2025

Dear Dr. Olsson,

Thank you for submitting your manuscript to PLOS ONE. After careful consideration, we feel that it has merit but does not fully meet PLOS ONE’s publication criteria as it currently stands. Therefore, we invite you to submit a revised version of the manuscript that addresses the points raised during the review process.

We look forward to receiving your revised manuscript.

Kind regards,

Anton Sokhan, Ph.D

Academic Editor

PLOS ONE

Journal Requirements:

Reviewers' comments:

Reviewer's Responses to Questions

**Comments to the Author**

Reviewer #3: All comments have been addressed

Reviewer #4: All comments have been addressed

Reviewer #5: All comments have been addressed

2. Is the manuscript technically sound, and do the data support the conclusions?

Reviewer #3: Partly

Reviewer #4: Yes

Reviewer #5: Partly

3. Has the statistical analysis been performed appropriately and rigorously?

Reviewer #3: Yes

Reviewer #4: Yes

Reviewer #5: I Don't Know

4. Have the authors made all data underlying the findings in their manuscript fully available?

Reviewer #3: Yes

Reviewer #4: Yes

Reviewer #5: No

5. Is the manuscript presented in an intelligible fashion and written in standard English?

Reviewer #3: Yes

Reviewer #4: Yes

Reviewer #5: Yes

Reviewer #3: I believe that I did not review the manuscript when submitted initially and this is my first review.

The most important problem inherent to the manuscript is the outcome, as stated in the previous reviews. The pathogens to cause BSI in ICU setting is well known issue and it is not unique even in Nordic countries. The manuscript did not show many, if any, new findings other than what we already know and the Nordic setting would not make this manuscript unique still.

Reviewer #4: (No Response)

Reviewer #5: This is a prospective, multicenter cohort of 354 invasively ventilated COVID-19 patients reporting a 17% rate of ICU-acquired bloodstream infection (BSI), pathogen distribution (dominance of S. aureus), and associations between ICU-acquired BSI and longer ventilation/LOS and higher 365-day mortality. The study adds useful Nordic data and benefits from prospective, multicentre case ascertainment and high blood-culture sampling rates. However, several important methodological, analytic and reporting issues must be addressed:

1. The current manuscript often reads as implying causality (“BSI increased mortality”). Because this is an observational cohort, causal claims must be tempered and rephrased as associations. The authors acknowledge this in the discussion, but the abstract and conclusion still overstate causality. More importantly, the temporal relationship between LOS/IMV duration and BSI is not rigorously handled. Patients who remain longer in ICU have more time at risk for BSI. The authors should perform time-to-event analyses such as Kaplan-Meier for time to first ICU-acquired BSI.

2. Blood cultures were drawn based on clinical judgement. What were the typical clinical indications? Was there a standardized sampling protocol across sites? This is necessary because differential testing can bias apparent incidence.

3. Provide a clear table listing all pathogens, counts, and percent of total BSIs.

4. Provide a table summarising empiric agents used (first-line and second-line), durations, and the proportion who received antibiotics prior to the positive blood culture.

5. Provide antimicrobial susceptibility patterns (resistance to 3rd-generation cephalosporins, MRSA, VRE, ESBL, carbapenem resistance) for the isolates. This affects interpretation of empiric therapy appropriateness and outcome.

6. BSIs in ICU are strongly associated with intravascular devices. The current dataset lacks device-level exposure variables (presence/duration of central venous catheters, number of line days, insertion practices, catheter care bundles), a limitation the authors note, but these data are critical. If such data are available in the records, they should be added and analyzed (CLABSI rates). If not available, the authors should, explicitly state this limitation(already partly done) and avoid implying modifiable device factors without data.

7. The multivariable model aiming to predict BSI at ICU admission has AUROC 0.66, which is poor. Authors should report how multicollinearity was assessed and the full list of variables considered (you mention p<0.2 rule but please provide a table of candidate variables).

**Do you want your identity to be public for this peer review?** For information about this choice, including consent withdrawal, please see our Privacy Policy

Reviewer #3: **Yes: ** kentaro iwata

Reviewer #4: No

Reviewer #5: **Yes: ** Associate Professor Khaoula Meddeb, Medical Intensive Care Unit, Farhat Hached University Hospital

---

## [Author Response · Author response to Decision Letter 2]

10 Dec 2025

We want to thank the editor and reviewers for their wise and insightful comments. We have tried our very best to make adequate changes to the manuscript and we have uploaded a new improved version of the manuscript as well as a copy of the manuscript with trackable highlighted changes. We believe that your comments have made it possible for us to substantially improve the quality of the manuscript.

Reviewer #3:

The most important problem inherent to the manuscript is the outcome, as stated in the previous reviews. The pathogens to cause BSI in ICU setting is well known issue and it is not unique even in Nordic countries. The manuscript did not show many, if any, new findings other than what we already know and the Nordic setting would not make this manuscript unique still.

Response: There is unclarity about the incidence of BSIs as well as the microbiological spectrum in the Nordic setting. Further, no consensus exists on factors that can predict an ICU-acquired BSI on ICU admission in critically ill COVID-19 patients. An ICU-acquired BSI is associated with worse outcome, and this manuscript adds to the existing research and emphasizes the importance of active vigilance on the development of an ICU-acquired BSI.

Reviewer #5:

1. The current manuscript often reads as implying causality (“BSI increased mortality”). Because this is an observational cohort, causal claims must be tempered and rephrased as associations. The authors acknowledge this in the discussion, but the abstract and conclusion still overstate causality. More importantly, the temporal relationship between LOS/IMV duration and BSI is not rigorously handled. Patients who remain longer in ICU have more time at risk for BSI. The authors should perform time-to-event analyses such as Kaplan-Meier for time to first ICU-acquired BSI.

Response: This is indeed an important remark. The primary aim of this study was to describe the aetiology and frequency of BSI. Secondary aims included associations between BSI and duration of IMV and LOS as well as all-cause mortality. We have clearly written throughout the manuscript, including the abstract, that the secondary objectives are mere associations and have not implied any causality from our findings. In line with reviewer 5’s remark, we have further toned down the association with mortality (Page 2, lines 55-56; page 18, lines 299-300; page 22, lines 402-404). We believe that further analyses, such as time-to-event analyses for time to first ICU-acquired BSI, would not bring us any closer to the possible causal relationship between BSI and mortality. This has been further clarified in the discussion (Page 20, lines 349-354).

2. Blood cultures were drawn based on clinical judgement. What were the typical clinical indications? Was there a standardized sampling protocol across sites? This is necessary because differential testing can bias apparent incidence.

Response: We have added additional information on regional guidelines for blood sampling in patients with a suspected severe infection or sepsis. The participating hospitals follow the guidelines from Surviving Sepsis Campaign Guidelines (Page 7, lines 135-143).

3. Provide a clear table listing all pathogens, counts, and percent of total BSIs.

Response: Please see supplementary table 1 for information regarding pathogens that were considered significant if they were identified in a blood culture and figure 2 for a presentation of how frequently a specific bacterial species was identified.

4. Provide a table summarising empiric agents used (first-line and second-line), durations, and the proportion who received antibiotics prior to the positive blood culture.

Response: Unfortunately, that specific data is not available in more detail than what is already presented.

5. Provide antimicrobial susceptibility patterns (resistance to 3rd-generation cephalosporins, MRSA, VRE, ESBL, carbapenem resistance) for the isolates. This affects interpretation of empiric therapy appropriateness and outcome.

Response: The situation regarding antimicrobial resistance in the Nordic countries has been reported to not be as severe as for example in the eastern and southern parts of Europe (see reference #21 in the manuscript). Hence, we consider it important to highlight the Nordic aspect of the study. Further, the low resistance decreases the need of specifying antibiotic resistance for specific species occurring in the study. Additionally, the rate of deaths associated with antimicrobial resistance is lower in the Nordic countries compared to most other countries according to a Lancet article published in 2024 (reference #36 in the manuscript). We acknowledge the insightfulness of the reviewer’s comment and since the first submission we have added information about the lower rate of antimicrobial resistance in the Nordic countries into the background section (Page 5, lines 90-95) as well as into the discussion (Page 20, lines 321-331).

In addition, we have checked the aggregated data from the intensive care register. There was a growth of multiresistent bacteria (MRSA, ESBL or VRE) in 0,9 % of the ICU admissions suffering from COVID-19 at the participating ICUs.

6. BSIs in ICU are strongly associated with intravascular devices. The current dataset lacks device-level exposure variables (presence/duration of central venous catheters, number of line days, insertion practices, catheter care bundles), a limitation the authors note, but these data are critical. If such data are available in the records, they should be added and analyzed (CLABSI rates). If not available, the authors should, explicitly state this limitation(already partly done) and avoid implying modifiable device factors without data.

Response: We have adjusted the manuscript in accordance with this response by adjusting the included tables and the manuscript.

7. The multivariable model aiming to predict BSI at ICU admission has AUROC 0.66, which is poor. Authors should report how multicollinearity was assessed and the full list of variables considered (you mention p<0.2 rule but please provide a table of candidate variables).

Response: Variables with a p<0.2 in the non-parametric analysis (see table 1) were included in the logistic regression. All variables with a p<0.2 in the non-parametric tests were analysed in univariable logistic regressions. Variables with a p<0,2 in the univariable logistic regression were included in the multivariable logistic regression. Please, see table 2 for data regarding the logistic regressions.

We have assessed multicollinearity by testing for strong correlations with Spearmans- and Point-Biserial-test and we could not identify any strong correlations (all correlations below 0,3) between any variables included in the multivariable regression analysis. We have now added this to the manuscript (Page 9-10, lines 188-191).

---

## [Decision Letter · Decision Letter 2]

22 Dec 2025

Aetiology and impact of bacterial bloodstream infections in mechanically ventilated COVID-19 patients: A prospective Swedish multicenter cohort study

PONE-D-25-02714R2

Dear Dr. Olsson,

We’re pleased to inform you that your manuscript has been judged scientifically suitable for publication and will be formally accepted for publication once it meets all outstanding technical requirements.

Kind regards,

Anton Sokhan, Ph.D

Academic Editor

PLOS One

Additional Editor Comments (optional):

Reviewers' comments:

Reviewer's Responses to Questions

**Comments to the Author**

Reviewer #3: All comments have been addressed

Reviewer #4: All comments have been addressed

2. Is the manuscript technically sound, and do the data support the conclusions?

Reviewer #3: Yes

Reviewer #4: Yes

3. Has the statistical analysis been performed appropriately and rigorously?

Reviewer #3: Yes

Reviewer #4: Yes

4. Have the authors made all data underlying the findings in their manuscript fully available?

Reviewer #3: Yes

Reviewer #4: Yes

5. Is the manuscript presented in an intelligible fashion and written in standard English?

Reviewer #3: Yes

Reviewer #4: Yes

Reviewer #3: (No Response)

Reviewer #4: (No Response)

**Do you want your identity to be public for this peer review?** For information about this choice, including consent withdrawal, please see our Privacy Policy

Reviewer #3: **Yes: ** kentaro iwata

Reviewer #4: No

---

## [Editor Report · Acceptance letter]

PONE-D-25-02714R2

PLOS One

Dear Dr. Olsson,

I'm pleased to inform you that your manuscript has been deemed suitable for publication in PLOS One. Congratulations! Your manuscript is now being handed over to our production team.

Kind regards,

on behalf of

Dr. Anton Sokhan

Academic Editor

PLOS One